# Plant Elongator—Protein Complex of Diverse Activities Regulates Growth, Development, and Immune Responses

**DOI:** 10.3390/ijms21186912

**Published:** 2020-09-22

**Authors:** Magdalena Jarosz, Mieke Van Lijsebettens, Magdalena Woloszynska

**Affiliations:** 1Department of Genetics, Faculty of Biology and Animal Breeding, Wroclaw University of Environmental and Life Sciences, ul. Kozuchowska 7, 51-631 Wroclaw, Poland; mjarosz48@gmail.com; 2Department of Plant Biotechnology and Bioinformatics, Ghent University, 9052 Ghent, Belgium; mieke.vanlijsebettens@psb.ugent.be; 3Center for Plant Systems Biology, VIB, 9052 Ghent, Belgium

**Keywords:** Elongator, transcription, translation, *Arabidopsis thaliana*, epigenetics

## Abstract

Contrary to the conserved Elongator composition in yeast, animals, and plants, molecular functions and catalytic activities of the complex remain controversial. Elongator was identified as a component of elongating RNA polymerase II holoenzyme in yeast, animals, and plants. Furthermore, it was suggested that Elonagtor facilitates elongation of transcription via histone acetyl transferase activity. Accordingly, phenotypes of *Arabidopsis elo* mutants, which show development, growth, or immune response defects, correlate with transcriptional downregulation and the decreased histone acetylation in the coding regions of crucial genes. Plant Elongator was also implicated in other processes: transcription and processing of miRNA, regulation of DNA replication by histone acetylation, and acetylation of alpha-tubulin. Moreover, tRNA modification, discovered first in yeast and confirmed in plants, was claimed as the main activity of Elongator, leading to specificity in translation that might also result indirectly in a deficiency in transcription. Heterologous overexpression of individual *Arabidopsis* Elongator subunits and their respective phenotypes suggest that single Elongator subunits might also have another function next to being a part of the complex. In this review, we shall present the experimental evidence of all molecular mechanisms and catalytic activities performed by Elongator in nucleus and cytoplasm of plant cells, which might explain how Elongator regulates growth, development, and immune responses.

## 1. Introduction

Elongator is a protein complex originally identified in yeast (*Saccharomyces cerevisiae*) when all six subunits ELP1-6 of the complex were found in a biochemical purification aimed at the elongating form of RNA polymerase II (RNAPII) [1]. The newly discovered complex was thought to bind and switch RNAPII from the initiation of transcription to the elongation state. The core subcomplex is formed by ELP1, ELP2, and ELP3, whereas the accessory subcomplex consists of ELP4, ELP5, and ELP6 [1,2,3]. A molecular architecture of the fully assembled Elongator complex reveals a symmetric dimer of ELP1, ELP2, and ELP3 with one of these subcomplexes bound to the heterohexameric ELP4, ELP5, ELP6 ring [4,5,6]. ELP3 is a catalytic subunit responsible for the enzymatic activity of the Elongator complex. It contains a C-terminal lysine (K) acetyltransferase (KAT) domain and an N-terminal radical S- adenosylmethionine domain (SAM). KAT activity was shown to be responsible for the acetylation of nucleosomal histone H3 lysine 14 and histone H4 lysine 8 in yeast, and therefore the domain was originally named histone acetyl transferase (HAT) [7]. However, later, the histone acetylation by this domain was challenged [8] and KAT was also described to target substrates other than histones which are tubulin [9,10,11] and tRNAs [8]. The N-terminal domain is a Fe_4_S_4_ cluster that is able to bind and cleave SAM in the reaction involved in tRNA wobble uridine modification [12]. ELP1 and ELP2 are primarily responsible for maintaining the structural integrity of the complex, whereas the accessory subcomplex binds tRNA molecules through ELP6 which allows the Elongator complex to perform tRNA wobble uridine modification [13].

Originally, facilitation of elongation during transcription was proposed to be the main role of yeast Elongator as indicated by the interaction between Elongator and the elongating form of RNAPII, and transcriptional defects in the *elp* mutants were observed [2,7]. Additionally, the Elongator complex is involved in exocytosis [14], telomeric gene silencing, and DNA repair [15]. Mutations in core Elongator subunits result in the resistance to the toxin zymocin [16,17], which is the RNAse enzyme targeting tRNA anticodons containing modified wobble uridine [18]. The mutant insensitivity to the toxin was convincingly explained by the Elongator activity in the tRNA wobble uridine modifications [12]; consequently, the Elongator-deficient *S. cerevisiae* mutants do not contain tRNAs targeted by zymocin. Due to the tRNA-related activity, the Elongator complex is involved in the fine-tuning control of protein translation. Deleterious *elp3* mutants are also characterized by growth defects such as sensitivity to temperature, salt, caffeine, and DNA-damaging factors [1,2,19].

Soon after Elongator was identified in yeast, successful isolation of the complex was also performed among multicellular eukaryotes like humans [20,21], *Mus musculus* [22], *Caenorhabditis elegans* [23], and *Arabidopsis thaliana* [24,25]. Interestingly, orthologues of the ELP3 protein are found in *archaea* wherein ELP3 catalyzes the tRNA wobble uridine modification [26,27]. The Elongator complex is highly conserved amongst eukaryotes both in terms of structure and interactions between subunits [28]. Cross species complementation analyses of genes encoding individual subunits and subdomains have experimentally proven Elongator’s conservation for yeast [29], insects [30,31], worms [23], plants [32], and humans [20].

Contrary to the well-preserved structure of Elongator in eukaryotes, activities of the complex and regulated processes are diverse and kingdom-specific. Human genes encoding Elongator subunits are associated with familial dysautonomia, which is a developmental disorder of the sensory and autonomic nervous system [33]. Defects in genes coding for Elongator subunits in mice lead to meiotic deficiencies during spermatogenesis and impairment of zygotic paternal genome demethylation in oocytes [34]. A single point mutation in *Elp6* results in neuron degeneration and ataxia-like behavior of *wobbly* mouse [35]. In *Drosophila melanogaster*, Elongator has been included in several processes such as larval- and neuro-development [30,31] as well as neurotransmitter release efficiency [36]. In mice and *C. elegans*, Elongator acetylates neuronal alpha tubulin [9,11]. In plants, Elongator is involved in growth, organ development [37,38], cell proliferation [24], cell cycle [39], immune response [40], abscisic acid (ABA), and stress responses [41,42]. *Elongata* (*elo*) mutants of *A. thaliana*, named for their elongated leaves, led to an identification of the Elongator subunits, and therefore, Elongator mutants in plants are designated *elo*/*elp*. The *elo*/*elp* mutants are characterized by features such as narrow leaves [24], defective root development [38], hypersensitivity to abscisic acid (ABA) [41], and defective skotomorphogenetic and photomorphogenetic development [43]. Plant growth [25,43], root development [38], and immune response [40,44] pathways are regulated during the transcription by the histone acetylation and/or DNA demethylation enzymatic activities of Elongator. Elongator mediates non-histone acetylation of alpha tubulin [10]; it is also engaged in the microRNA biogenesis [45,46]. Finally, plant Elongator takes part in controlling the translation through its activity in tRNA wobble uridine modification [32,47,48,49,50]. Highly conserved structure of Elongator within eukaryotes and its kingdom-specific roles lead to a conclusion that Elongator regulates processes in growth and development, and response to different stimuli.

Initially, genetic data supported Elongator’s role in the transcription. However, a shift in the way of understanding the Elongator’s role took place when genetic deletion of ELP3 *Schizosaccharomyces pombe* ortholog resulted in a reduction of the conserved modification of uridine in the wobble position in the tRNA anticodon loop. As for *Saccharomyces cerevisiae*, Elongator’s role in multiple tRNA modifications was also confirmed [12]. In humans, Elongator was also found to be involved in processes other than those connected with the nucleus as Elongator subunits were located primarily in cytoplasm [20,51,52,53]. Final evidence was provided by the fact that all phenotypes of Elongator-deficient yeast cells are linked to tRNA modification and not transcription, exocytosis, telomeric gene silencing, or DNA repair [19,54]. Thus, defects observed in yeast *elp* mutants are caused by an indirect effect of deficient tRNA wobble modification which increases the decoding efficiency of the A- and G-ending codons [19,55,56,57]. Therefore, Elongator-mediated tRNA modification fine-tunes translation of individual proteins, and the protein synthesis is affected differently depending on the amino acid content [32].

Elongator’s involvement in tRNA modification appears to be evolutionary conserved. Mutations in genes encoding Elongator subunits cause deprivation of tRNA modification in *C. elegans* [23] and *A. thaliana* [32]. Similarly, in mouse germ cells with ELP1-deficiency, wobble uridine modification was found at a lower level [34]. The *elp6* mutation in *wobbly* mouse destabilizes the accessory subcomplex, leading to lower levels of tRNA modifications and defects in protein translation fidelity and folding [35]. Furthermore, brain tissues and fibroblasts isolated from patients with familial dysautonomia also had lower levels of tRNA modification [58,59]. Therefore, in addition to structure, Elongator-mediated tRNA modification is also conserved in eukaryotes. As shown in bacterial *Dehalococcoides mccartyi* DmcELP3, anticodon stem loops of tRNA molecules are bound by a cleft formed by KAT and SAM domains, and therefore allowing ELP3 to perform the carboxy-methyl (cm^5^) modification, and one of the steps of this reaction includes hydrolysis of acetyl-CoA catalyzed by the KAT domain [60]. In 2019, Lin and co-workers studied structural and catalytic properties of ELP3, and identified the subunit as a non-canonical acetyltransferase and showed that specific tRNAs, but neither histone or tubulin peptides nor other nucleic acids, act as an exclusive trigger for acetyl-CoA hydrolysis in bacterial, archaeal, and eukaryotic (yeast) ELP3s [8]. With this pool of recent data, it is now believed that Elongator’s primary or sole function in yeast is to modify wobble uridines of tRNA, and that transcription defects observed in Elongator mutants constitute a consequence of a deficiency in tRNA modification which was shown to regulate translation [57]. On the other hand, studies of plant Elongator provide a wide spectrum of physiological processes affected in *elo*/*elp* mutants and indicate that Elongator regulates diverse molecular mechanisms through histone and non-histone protein acetylation, DNA (de)methylation, tRNA modification, and potential unknown activities involved in microRNA biogenesis. Furthermore, histone acetylation and DNA (de)methylation are linked to changes in the expression of specific genes related to physiological and molecular processes which are affected by the loss of Elongator which strongly supports its role in the transcription regulation. Therefore, the intriguing and still unanswered question about plant Elongator is whether different biochemical activities of the complex are synchronized in regulating individual physiological processes.

The major controversy concerns the true biochemical role of plant Elongator and molecular mechanisms regulated directly versus indirectly by the complex. Another unclear issue is whether Elongator always acts as a complex, or are there any activities fulfilled by individual subunits or only by one of two subcomplexes? A large amount of evidence indicates that Elongator works as a complex because in yeast, animals, and plants, similar results were obtained for mutants carrying deleterious mutations in genes encoding diverse subunits of Elongator, which indicates that the complex as a whole regulates diverse processes, for example, skoto- and photomorphogenesis [42]. However, recent data from overexpression experiments on species other than *Arabidopsis* suggests a possible role for single subunits. In this review, we shall focus on studies on plant Elongator which concern these two main controversies.

## 2. In Plants Elongator is Located and Active in Nucleus and in Cytoplasm

In the last ten years, subcellular localization of the Elongator complex has been investigated in diverse plant tissues, mainly by expressing various ELP subunits fused to fluorescent reporter proteins in the respective *elo*/*elp* mutants and analyzed by means of confocal microscopy and/or immunological detection. Functionality of fusion proteins was usually confirmed by their ability to rescue phenotypes of mutant plants. In 2010, Nelissen and co-workers showed that the GFP-ELO3 fusion protein was detected predominantly in the nucleus and, to a lesser extent, in the cytoplasm of primary root cells in young Arabidopsis thaliana seedlings [25]. The dominant nuclear and partly cytoplasmic localization of Elongator was supported two years later by Tran and co-workers who performed confocal laser scanning microscopy to detect the red fluorescent protein tagged ELP3 in guard cells of *Vicia faba* epidermal leaf cells [10]. Elongator interacted with transcript elongation factors in the affinity purification and mass spectrometry assays confirming nuclear localization of the complex in suspension cultured cells of *Arabidopsis thaliana* [61]. Using the bimolecular fluorescence complementation (BiFC) assay, subunits of Elongator were shown to interact with the Dicing complex components in nuclear bodies [45]. ELP6 subunit expressed in fusion with yellow fluorescent protein VENUS and under its own autologous promoter was detected in the cytoplasm of *Arabidopsis thaliana* root meristem stele cells, while in the *35S::YFP:ELP3* overexpression line, both cytoplasmic and nuclear localization was observed [48]. In the most recent report, Qi and co-workers identified ELP3-GFP expressed from the autologous promoter in stele meristem cells, predominantly in the cytoplasm and to lesser extent in the nucleus [62]. In the case of the *p35S:GFP-ELP3* line, nuclear localization was more evident in the columella and epidermis cells in the root elongation zone. Finally, cell fractionation approach combined with immunodetection of endogenous ELP1 and ELP3 proved that Arabidopsis Elongator is located in both the cytoplasm and the nucleus. To prove that Elongator is indeed active in the nucleus, ELP3-GFP was fused with SV40 nuclear localization sequence, the resulting protein was detected in the nucleus and shown to fully rescue root growth defects of *elp3* mutants [62].

In conclusion, the vast majority of reports confirm both nuclear and cytoplasmic localization of Elongator, indicating two separated pools of the complex coexisting in the cell. Considering the aforementioned, the complex might perform diverse activities specific for the two compartments.

## 3. Elongator in Nucleus

The plant Elongator complex located in the cell nucleus is considered to regulate transcription of genes encoding proteins at the elongation stage and via DNA (de)methylation as well as pri-miRNA transcription and processing. Additionally, nuclear Elongator is also involved in DNA replication and cell-cycle progression.

### 3.1. Plant Elongator Regulates Transcription

#### 3.1.1. Processes, Mechanisms, and Pathways Regulated by Elongator via Control of Transcription

Elongator regulates plant development at various stages starting from germination described as severely retarded in *elo3* mutants [24]. The detailed systematic study, wherein germination is defined as the time point when endosperm rupture occurs in more than 50% of seeds, showed that in darkness, *elo3-6* mutant germinates 6 h later than wild type Col-0 [63]. Further analysis of skoto- and photomorphogenesis, i.e., early seedling development in darkness and light, respectively, indicated that the early post-germination growth was also regulated by the Elongator complex [43,63]. In darkness, hypocotyls of *elo/elp* seedlings did not elongate as rapidly as wild type Col-0, while during photomorphogenesis, *elo/elp* seedlings were hyposensitive to red, far-red, and blue light as evidenced by longer hypocotyls and hyponastic cotyledons. The best-known feature of *elo*/*elp* mutants depleted in Elongator subunits is narrow and elongated leaf shape. This phenotype was later explained by a decreased palisade cell number [24], indicating that Elongator stimulates cell proliferation. Another phenotypic characteristic observed in *elo* mutants is the shorter primary root resulting from a reduced cell number and length pointing to the role of Elongator in both cell proliferation and cell elongation [24,38,62]. The meristem zone was smaller with a decreased cell number and also the elongation zone was shorter due to a reduced cell length [38]. Elongator is also required to maintain cell identity in the root stem cell niche [38,62] and to regulate radial patterning in roots [62]. Root phenotype together with the reduced apical dominance, abnormal inflorescence phyllotaxis, and open leaf venation observed in *elo/elp* mutants are all signs of defective auxin signaling and distribution [25]. The crucial role of the Elongator complex during embryogenesis and seed development was evidenced by synergistic embryo lethal phenotype of double mutants combining *elo1* or *elo3* and *hub1* mutation in gene encoding HISTONE MONOUBIQUITINATION 1–enzyme facilitating elongation of transcription [64]. The co-expression of *ELO3* and *HUB1* at the torpedo stage, when double mutants’ embryos arrested, was shown by in situ hybridization and confirmed that cooperation of Elongator and HUB1 is essential for gene expression during seed development.

Apart from the important role played by Elongator in plant development, the complex has also been confirmed to regulate several response processes at the transcriptional level. Disruption of Elongator results in hypersensitivity to abscisic acid, a plant hormone regulating seed maturation, plant growth, stress responses; it also plays an important role in triggering stomatal closure. Accordingly, ABA hypersensitivity in *elo*/*elp* mutants affected seedling growth and promoted stomatal closing; on the other hand, mutants were more resistant to oxidative stress, they were also more drought-tolerant and had an increased accumulation of anthocyanins [41,42]. What is particularly well studied is the function which Elongator had in various types of immune responses which belong mainly to plant host resistance. Mutations in *ELO* genes impaired immune mechanisms relaying on rapid transcriptional changes, and identified basal resistance and effector-triggered immunity as regulated by Elongator [40,44,65,66]. More recently, the Elongator complex has also been presented as an important regulator of the non-host resistance against bacterial pathogens [67].

Taken together, Elongator active as a regulator of transcription is involved in a wide range of developmental processes, from germination to vegetative, generative, and seed production phases; it is also essential during stress and immune responses.

#### 3.1.2. Transcript Levels of Individual Genes and Whole Transcriptomes are Altered in elo/elp Mutants and Plants Overexpressing Elongator Subunits

The discovery of a development or response-related defects in *elo/elp* mutants was followed by assays aimed at the detection of individual differentially expressed genes or transcriptome analyses, which could reveal gene expression alterations underlying the observed mutant phenotypes. The cDNA-AFLP and microarray experiments—performed to compare five *elo/elp* mutants to wild type—showed that genes differentially expressed in the mutants clustered together and all five mutations affect the transcription similarly [24]. As Elongator positively regulates the transcription, the downregulated genes were analyzed revealing the overrepresented Gene Ontology (GO) categories of chromatin assembly, pattern specification, vascular tissue development, and response to auxin stimulus correlating with the developmental defects of *elo/elp* mutants related to deficient auxin signaling and distribution [25]. Delayed germination of the *elo3-6* mutant was linked to 9 germination-related genes identified as downregulated in the microarray data comparing gene expression in darkness grown mutant and wild type seedlings [63]. Analysis of the same dataset allowed to identify the downregulated GOs which were assembled into the growth-controlling network consisting of four main hubs: circadian clock, regulators of skoto- and photomorphogenesis, hormone pathways, and the primary and the secondary cell wall biogenesis [43]. The network connects upstream regulators and downstream growth effectors via signal transmission pathways. Downregulation of components of this network explains the delayed hypocotyl elongation of mutant seedlings grown in darkness. On the other hand, young *elo3-6* and Col-0 seedlings grown in various light qualities and surveyed by the qPCR assay revealed positive photomorphogenesis factors to be downregulated in a mutant, which is in line with disturbed photomorphogenesis [43]. In *elp2* mutant, the defective root stem cell niche and quiescent center maintenance were associated with a decreased expression of transcription factors crucial for cell identity: *PLT1* (*PLETHORA 1*), *PLT2* (*PLETHORA 2*), *SCR* (*SCARECROW*), *SHR* (*SHORT ROOT*), *WOX5* (*WUSCHEL-RELATED HOMEOBOX 5*), and the auxin efflux transporter *PIN1* [38]. Downregulated expression of transcription factors was confirmed by means of qPCR, and additionally, by the analysis of transgenic lines transformed with constructs containing reporter fluorescent protein fused to promoter, or promoter and coding sequence of a given transcription factor. Low expression of *SHR* was shown by similar methods and by in situ hybridization in another *elo/elp* mutant—*elo1*, and supported by results of qPCR assay revealing decreased levels of *SHR* transcriptional targets including *SCR* [62].The increased resistance of *elo/elp* mutants to oxidative [42] and drought [41] stress is very likely associated with higher accumulation of anthocyanins in the mutants, resulting from much lower expression of the *MYBL2* gene encoding negative regulator of anthocyanins biosynthesis [42].

Positive regulation of immune response by Elongator was concluded based on defense deficiencies in *elo/elp* mutants which correlated with the aberrant expression of important defense genes [40,44,65,66,67,68]. During the bacterial pathogen induced defense response in five *elp1*-*elp5* mutants, transcript levels of the early defense genes *FRK1 (FLG22-INDUCED RECEPTOR-LIKE KINASE1*), *WRKY29*, and *GST1 (GLUTATHIONE S-TRANSFERASE1*) were reduced, while the expression of the late defense *PR* (pathogenesis-related) genes was blocked [67]. Following the induction of defense response with exogenous NAD^+^, expression of *PR* genes was either severely delayed (*PR1*) or almost completely inhibited (*PR2* and *PR5*) in the mutants [68]. Inhibition of defense response to bacterial pathogen observed in *elo/elp* mutants involved the decreased accumulation of reactive oxygen species (ROS) and salicylic acid (SA), and was in line with the suppressed expression of ROS production pathway gene *AtrbohD* (*Arabidopsis respiratory burst oxidase homologue*) and SA biosynthesis gene *ICS1 (ISOCHORISMATE SYNTHASE1*) [67]. The time-course gene expression profiles-performed by means of Northern blotting, qPCR, or microarray-revealed that induction of many defense genes is not only decreased, but also postponed in *elp2* or *elp3* mutants and may increase with significant delay to reach the level of the wild-type or lower [40,44,65,66]. Therefore, the authors have shown that Elongator regulates the kinetics of the rapid defense gene induction during basal, and more importantly, during effector-triggered immunity which is a highly-accelerated and amplified form of the immune response in plants. The kinetics of large scale transcriptome reprogramming, analyzed by the microarray technology, was slower in *elo2* mutant compared to the wild-type after being infected with both bacterial [40] and fungal [44] pathogens. The delayed expression was also confirmed for several individual and particularly important defense genes: *PR1* [65], *PR2*, *PR5*, *ICS1*, *WRKY18*, *WRKY33*, *EDS1* (*ENHANCED DISEASE SUSCEPTIBILITY1*), *EDS5*/*SA INDUCTION DEFICIENT1*, *PAD4* (*PHYTOALEXIN DEFICIENT4*), *NPR1* (*NONEXPRESSOR OF PATHOGENESIS*-*RELATED GENES1*) [66], *NPR1*, *EDS1*, *EDS5*, *PAD4*, *NDR1* (*NON*-*RACE*-*SPECIFIC DISEASE RESISTANCE1*), *ALD1* (*AGD2*-*LIKEDEFENSE RESPONSE PROTEIN1*) [40], *ORA59* (*OCTADECANOID-RESPONSIVE ARABIDOPSIS AP2*/*ERF59*), and *PDF1.2* (*PLANT DEFENSISN1.2*) [44].

Contrary to numerous reports confirming the decreased gene expression in the Elongator-deficient mutants, experiments demonstrating the upregulation of the respective genes in response to overexpression of the ELP subunits, are less common but similarly convincing. Heterologous overexpression of the *Arabidopsis thaliana* ELP genes in strawberry *Fragaria vesca* L. and in tomato *Solanum lycopersicum* caused increased constitutive or pathogen induced defense gene expression [69,70].

#### 3.1.3. Elongator Regulates Elongation of Transcription

The co-purification of the Elongator complex with the phosphorylated (elongating) RNAPII in yeast [1] was an early indication that Elongator regulates transcription during the elongation phase, most likely via its histone acetylation activity. Several lines of evidence prove that the transcription elongation related function of Elongator is conserved in plants, including interaction of the complex with RNAPII and transcript elongation factors [25,61,62,71], the role in RNAPII recruitment and mutant sensitivity to transcription inhibitor [62], and finally, binding to the coding regions of the regulated genes [44] and histone modifications [25,38,40,43,44,67] and/or DNA (de)methylation [38,40]. The immunocolocalization of the *Arabidopsis thaliana* ELP3 subunit with the elongating form of RNAPII and euchromatin associated histone marks was described already in 2010 [25]. The direct interaction between the *Arabidopsis* ELP4 and ELP5 subunits and the C-terminal domain (CDT) of RNAPII was detected recently in yeast two-hybrid assays and confirmed in planta by the firefly luciferase complementation imaging (LUI) in the *Nicotiana benthamiana* leaves [62]. Antosz and co-workers analyzed the RNAPII Transcript Elongation Complex (TEC) in *Arabidopsis* suspension cultured cells using reciprocal protein tagging combined with affinity purification and mass spectrometry, and identified Elongator among proteins repeatedly associated with TEC factors SPT4 (suppressor of Ty 4), TFIIS, and PAF1-C (RNAPII-associated factor 1 complex) and to a lesser extent with RNAPII [61]. The association between ELP1 and the transcript elongation factor SPT4 was confirmed by coimmunoprecipitation of the two proteins in the protein extract of young *Arabidopsis* seedlings and by the LUI assay in *N. bentamiana* cells [62]. Interaction and cooperation of Elongator and SPT4/SPT5 during transcription elongation is suggested by similar phenotypes of *elo/elp* mutants and the SPT4-RNAi plants showing a reduced leaf and root cell proliferation and defective venation, as well as common downregulated auxin-related genes [72]. Elongator interacts also physically and genetically with another transcription elongation factor IYO which bounds RNAPII in in vitro pull-down and in vivo reconstitution assays [71].

The importance of Elongator in RNAPII recruitment during transcription elongation was confirmed by significantly lower in *elp1* versus wild-type accumulation of RNAPII on the transcription start site and, to the greater extent, on the coding region of the *SHR* gene [62]. In the same study, *elp1* mutants were more resistant to 6-azauracil used as an inhibitor of transcription elongation, which additionally corroborates the role of Elongator during the elongation of transcription.

Elongator epigenetically facilitates the transcription elongation via the reaction catalyzed by the KAT domain of ELP3 resulting in acetylation of the histone H3 lysine-14 residues (H3K14Ac) in coding regions of a gene [7]. This activity of Elongator in plants was confirmed by a slightly decreased acetylation level of the H3K14 in *elp1* and *elp3* mutants [62], but mainly by the reduced histone acetylation in coding regions of individual genes. Elongator modifies histone acetylation with high selectivity (reviewed by [73]) and the number of plant genes identified as regulated by Elongator via histone acetylation is restricted to twenty (Table 1). To find genes targeted by Elongator for histone acetylation, candidates with transcript levels decreased in *elo/elp* mutant are usually selected based on their role in the pathway(s) affected by the *elo* mutation. Histone acetylation in the candidate gene is compared between the mutant and the wild-type by means of the Chromatin Immunoprecipitation followed by qPCR (ChIP-qPCR). Chromatin is precipitated with antibodies against H3K14Ac and used to isolate DNA which is quantified in qPCR with primers designed to amplify the promoter and the coding regions of the candidate gene. Reduced histone acetylation in the *elo3-6* mutant was detected in the auxin repressor gene *SHY2* (*SHORT HYPOCOTYL2*) and auxin influx carrier gene *LAX2* (*AUXIN TRANSPORTR*-*LIKE PROTEIN2*)—the two first Elongator targets involved in plant growth regulation via auxin signaling and transport, and therefore corresponding to the auxin-related phenotype of *elo/elp* mutants [25]. The analysis of the hypocotyl elongation delay of the darkness grown *elo3-6* seedlings led to the identification of genes coding for three growth regulators and the highest order transcription factors as direct targets of the Elongator KAT activity: *LHY* (*LATE ELONGATED HYPOCOTYL*), *HYH* (*HY-5 HOMOLOG*), and *HFR1* (*LONG HYPOCOTYL IN FAR-RED 1*) [43]. Genes of important transcription factors controlling root development: *PLT1*, *PLT2*, *SHR*, *SCR*, and gene of auxin efflux transporter *PIN1* were recognized as targeted by Elongator for histone acetylation in line with their decreased expression and root defects of the *elp2* mutant [38]. The *SHR* gene was also confirmed as the Elongator target by Qi and co-workers [62]. The largest group of the known Elongator KAT activity targets belongs to the defense pathways and comprises ten genes: *NPR1*, *PAD4*, *EDS1*, *PR2*, *PR5* [40], *WRKY33*, *ORA59*, *PDF1.2* [44], *AtrbohD*, and *ICS1* [67]. Five of these genes (*WRKY33*, *ORA59*, *PDF1.2*, *AtrbohD*, and *ICS1*) were identified as targeted by Elongator not only based on the histone acetylation levels reduced in the *elp2* mutant, but also through direct association of ELP2-GFP with the chromatin of the genes detected in the ChIP-qPCR assays using anti-GFP antibodies [44,67]. Only in the case of the *WRKY33* gene, the association was identified exclusively in the coding region, while for the remaining genes, the chromatin of both promoters and the coding regions associated with ELP2-GFP. Apparently, Elongator may bind to the promoters and the coding regions of the targeted genes, however, its KAT activity is switched on when it moves—possibly as the component of the transcript elongation complex—to the coding part of the gene.

#### 3.1.4. Elongator Promotes Biogenesis of miRNAs

Regulation of the transcription elongation via histone acetylation is the best-known nucleus- and transcription-related role of Elongator played in expression of the protein-coding genes. However, the Elongator complex is also involved in expression of the miRNA-coding genes and is essential for both transcription and processing of the pri-miRNA molecules [45] (Figure 1, Inset 1). The ChIP assay indicated that Elongator is required for the full association between RNAPII and transcribed *MIR* genes, and therefore positively regulates their transcription as confirmed by reduced accumulation of some miRNAs in *elp1*–*elp6* mutants coinciding with the increased levels of their complementary target transcripts. Resulting thereof deregulated expression of genes controlled by miRNAs, including factors involved in leaf development and auxin responses, might contribute to the most characteristic features of *elo*/*elp* mutants (reviewed by [73]). Elongator also interacts with the DCL1 (Dicer-like 1), HYL1 (HYPONASTIC LEAVES 1), and SE (SERRATE) factors of the Dicing complex in Dicing bodies at the nucleus [45]. Moreover, Elongator is crucial for DCL1 association with chromatin, enabling the co-transcriptional processing of pri-miRNAs by DCL1. As the plant Elongator is known to interact with RNAPII, the dual role of the complex during miRNA biogenesis is proposed to rely on simultaneous interactions with RNAPII and DCL1. These interactions physically and functionally tie together the transcription and processing of pri-miRNAs.

#### 3.1.5. Elongator Modifies DNA Methylation

The radical SAM domain of the ELP3 subunit was originally considered active in histone demethylation; however, its role in paternal DNA demethylation in mice zygotes was described later [22]. In plants, changes in DNA methylation were analyzed in *elo*/*elp* mutants of *A. thaliana* in the context of immune response [40] and root development [38]. Methylation levels of cytosines in promoters or the coding regions of several individual genes were compared between the *elp2* mutant and wild type by means of bisulfite sequencing. Methylation of cytosines was either increased (*PAD4*) [40] or reduced (*CYCB1*, *B-type CYCLIN1*) [38], or conversely altered in promoter and coding region of *NPR1* [40] in the *elp2* mutant. The opposite changes in cytosine methylation in the *elp2* mutant in different genes cannot be explained by DNA (de)methylation activity of Elongator. Genome-wide bisulfite deep sequencing also resulted in complicated patterns of cytosine methylation, showing that in the *elp2* genome, the total number of methylated cytosines was higher than in wild type, however, the average methylation levels of cytosines were lower [40]. Therefore, the significant differences in cytosine methylation patterns between the *elp2* mutant and wild type indicate that Elongator is somehow involved in shaping the methylation landscape of the genome (Figure 1, Inset 1), but the exact mode of action of the complex is elusive. In a tomato line with silenced *ELP2* gene, the gene expression of the DRM7, DRM8, and MET1 methyltransferases was enhanced which may suggest that Elongator affects DNA methylation indirectly [74]. Moreover, SAM-dependent DNA methyl transferases differ structurally from radical SAM enzymes such as the ELP3 subunit of the Elongator complex. Hence, it seems unlikely that the Elongator complex is able to catalyze both methylation and DNA demethylation in addition to its proven radical SAM activity.

### 3.2. Elongator Regulates Replication and Cell-Cycle Progression

The majority of research articles concerning the nuclear functions of the plant Elongator is focused on the transcription-related activity of the complex, whereas its role in DNA replication is less recognized. In 2011, Xu and co-workers published a comprehensive study explaining how the KAT activity of Elongator is involved in DNA replication, and how aberrant replication in *elo*/*elp* mutants leads to defective cell cycle progression [39]. The replication defect was suggested by a slower increase of the nuclear ploidy in the *elo3* mutant compared to wild type and reduced incorporation of EdU (5-ethynyl-2′-deoxyuridine), reflecting the rate of active DNA synthesis. In addition, the pull-down, co-immunoprecipitation, and ChIP experiments in planta indicated physical interaction of Elongator and PCNA (proliferating cells nuclear antigen), an important replication factor, and their possible common association with replicons during DNA replication. Reduced levels of the H3K56Ac and H4K5Ac histone acetylation signals in the *elo3* nuclei, specifically in replicons chromatin, implied that Elongator is required for DNA replication-coupled H3 and H4 acetylation (Figure 1, Inset 2). The aberrant replication most possibly caused the inefficient DNA repair because accumulation of damaged DNA molecules was very high in the *elo3* mutant and led to activation of DNA replication checkpoint to arrest the cell cycle.

The crucial role played by Elongator in DNA replication was confirmed [75], showing that mutation in the *ELO3* gene leads to the failure in DNA replication and activation of cell division in the meristem following germination, which results in blocked cell cycle and arrested seedling growth. Interestingly, in situ hybridization detected dramatically reduced histone H4 mRNA levels in the shoot apex and meristem root zone of the *elo3-14* mutant corresponding to aberrant DNA replication. Similarly, the transcript levels of several cell cycle-related genes, estimated using the semi-quantitative RT-PCR assay, were downregulated in line with arrested cell cycle. Therefore, it is possible that Elongator regulated progression of the cell cycle via replication-coupled histone acetylation and one of its’ transcription-related functions.

## 4. Elongator in Cytoplasm

Elongator complex in cytoplasm is required for post-transcriptional regulation of PIN auxin transport proteins, endoreduplication cycling, and possibly for drought stress response via tRNA wobble uridine modification. Secondly, Elongator acetylates alpha tubulin and through this process, Elongator is involved in the regulation of microtubules dynamics.

### 4.1. Processes and Mechanisms Regulated by Elongator via Activity in the Cytoplasm

Apart from cell nuclei, the Elongator complex is also active in the cytoplasm wherein this protein is involved in tRNA wobble uridine modification and acetylation of alpha tubulin. tRNA modification activity fine-tunes the translation process, and therefore Elongator is an important regulator of protein synthesis in plants. In the cytoplasm, Elongator also acetylates alpha tubulin and thus it is involved in regulating microtubules dynamics. Elongator-deficient plants both at the seedling phase and later developmental stages show defects in patterning, and shoot and root morphology [48,76]. It was found that these abnormalities are a consequence of the lack of the tRNA wobble uridine modification, leading to a reduced abundance of PIN auxin transport proteins. Therefore, the impaired regulation of translation leads to a defective auxin distribution. The second Elongator-dependent developmental process linked to its function in translation is leaf morphogenesis. tRNAs with wobble uridines modified by the Elongator complex are important for endoreduplication cycling, which represents a crucial process for proper formation of epidermal and mesophyll tissues [50]. Another interesting finding about Elongator’s activity in translational regulation is a strong resemblance between *elo3*-*6* and *grxs17* mutants at the morphological, molecular, and physiological level. These similarities indicate corresponding roles of Elongator and GRXS17 proteins in tRNA modification [49]. Moreover, recent studies in poplar (*Populus trichocarpa)* indicate that tRNA modification activity of the Elongator complex requires interaction with the protein products of the *PtKTI12* genes—the yeast Kti12 ortholog required for stress response and drought stress tolerance [77].

Taken together, Elongator-mediated tRNA modification is required for fine-tuning the translation of individual proteins by increasing the decoding efficiency. Thus far, only a few processes were linked to the plant Elongator’s activity in the cytoplasm which are auxin responses and endoreduplication cycling—crucial for proper leaf morphogenesis. Most probably there are more processes controlled by the Elongator complex tRNA modification activity which are not defined yet.

### 4.2. Elongator Modifies Wobble U_34_ in tRNAs and Regulates Protein Translation

#### 4.2.1. Conservation of Wobble U_34_ Modification between Yeast and Plants

In plants, Elongator’s activity during the translation was firstly described when *Arabidopsis* mutants lacking the ELP3 subunit showed defects in tRNA wobble uridine modification, which is at the 34th position around the anticodon stem loop [32] (Figure 1, Inset 3). tRNA modifications of bases located in this region enhance the decoding potential and accuracy of codon-anticodon pairing [78,79], which is provided by creating chemical bonds between the anticodon stem loop and its cognate and near-cognate codons [80,81]. This type of tRNA modifications primarily occurs at the 34th and 37th positions around the anticodon stem loop [79,81,82], and they are important for fine-tuning the elongation phase during protein synthesis [83,84]. Mehlgarten and co-workers were the first to show that the plant Elongator is involved in the mcm^5^s^2^U_34_ (5-methoxycarbonylmethyl uridine) and ncm^5^U_34_ (5-carbamoylmethyl uridine) tRNA modifications [32]. It is proposed that the Elongator complex catalyzes the attachment of carbonylmethyl to the 5th position of the tRNAs wobble uridine which results in forming 5-carbonylmethyl uridine-cm^5^U_34_ [27,32]. The wobble uridine after modification by the Elongator complex may be converted into ncm^5^U_34_ through an unknown mechanism or further methylated and thiolated to create mcm^5^s^2^U_34_, wherein the wobble uridine has additional sulfur atom incorporated into the 2nd position. In plants, the tRNAs transporting lysine-LysUUU, glutamic acid-GluUUC, and glutamine GlnUUG are targeted for thiolation [49,85,86].

#### 4.2.2. The Role of the DRL1 Interactor of Elongator in tRNA Modification

In yeast, direct contact with motifs of the killer toxin-insensitive12 protein (Kti12) seems to be required for the Elongator-mediated tRNA modification [87,88]. In *Arabidopsis*, deformed root and leaves1 protein (DRL1) represents an orthologue of Kti12 [16,89,90]. Jun and co-workers performed functional comparative analysis of the yeast Kti12 and Arabidopsis DRL1, indicating low overall homology between these proteins; however, the presence of structurally conserved domains in different species was noted in DRL1 [91]. Complementation experiments in the yeast *kti12* with *Arabidopsis* and rice *DRL1* rescued growth retardation suggesting conserved function [91]. However, the function of DRL1 in caffeine sensitivity and zymocin-mediated growth inhibition did not overlap with Kti12 [91]. Later, the lack of functional exchange was additionally confirmed, suggesting the evolutionary diversification of Elongator and its regulatory proteins [87]. DRL1 is physically associated with the Elongator complex [92] and the yeast Kti12 motifs required for cofactors binding were shown to be conserved in the plant DRL1 [87], suggesting that similarly to Kti12, DRL1 is involved in the Elongator activity in translational regulation. In *Arabidopsis*, similar morphological phenotypes of *drl1* and *elo*/*elp* mutants together with overlapping transcriptome changes seem to confirm corresponding functions of DRL1 and the Elongator complex [24,39,90]. Furthermore, DRL1 is involved in processes that are also regulated by the Elongator complex, such as cell division and differentiation, establishment of adaxial-abaxial polarity [91], and contribution to defense responses; however, Elongator has a broader role in plant immunity [92]. Moreover, DRL1 is required for the ncm^5^U_34_ modification which represents one of the tRNA wobble uridine modifications [47]. The newest findings in poplar—a woody plant phylogenetically close to *Arabidopsis*—showed that expression of *PtKTI12A* and *PtKTI12B* was induced by drought and heat stresses, whereas downregulation of these genes increased the drought tolerance [77]. Additionally, reduced levels of mcm^5^s^2^U_34_, mcm^5^s^2^U_34_, ncm^5^U_34_ modifications were identified by means of liquid chromatography–mass spectrometry LC-MS [92]. Moreover, similarly to the Elongator complex, PtKTI12A and PtKTI12B proteins are located both in the nucleus and the cytoplasm. Therefore, Wang and co-workers speculate that in woody plants, Kti12 homologs interact with the Elongator complex to participate in tRNA wobble uridine modifications [77]. Recently, Krutyholowa and co-workers presented the crystal structure of *Chaetomium thermophilum* N-terminal domain of Kti12 in transition state of ATP hydrolysis, and its structure strikingly resembles PSTK-an archaea O-phosphoseryl-tRNA kinase required for the synthesis of tRNA molecules transferring selenocysteine tRNASec [93]. The architecture of CtKti12 nucleotide binding pocket resembles conserved canonical P-loop ATPases. The P-loop present in Kti12 proteins is probably involved in ATP binding and hydrolysis, important for the Elongator complex to bind tRNAs [93]. Predicted structure of *Populus* Kti12 showed overall resemblance to CtKti12, including loop regions, suggesting that in plants, Kti12 is involved in tRNA binding and wobble uridine modification [77].

#### 4.2.3. The role of Elongator-Mediated tRNA Modification in Auxin Responses

Further research on the Elongator’s role in the control of translation showed that its tRNA modification activity takes part in auxin distribution and responses. Mutations in the Elongator subunits ELP3 and ELP6 cause defects in the auxin-controlled development which is associated with the reduced abundance of PIN-formed (PIN) auxin transport proteins [48]. No changes in *PIN* transcript level were observed, suggesting that Elongator’s tRNA modification activity is involved in the post-transcriptional PINs regulation. Indeed, a reduction in mcm^5^s^2^U_34_ and ncm^5^s^2^U_34_ modifications was observed in a high-pressure liquid chromatography analysis. Moreover, expression of the endonuclease γ-toxin under *RP40* promoter (*RP40p::gam*) caused the auxin-related defects in *Arabidopsis* resembling the phenotype of the Elongator-deficient plants [48]. γ-toxin preferentially degrades tRNAs modified by Elongator [18], and accordingly, the reduced abundance of tRNAGlnUUG and tRNAGluUUC was detected in *RP40p::gam*. Similar plant auxin-related abnormalities were observed after the downregulation of the *RNA ligase RNL* involved in tRNA splicing [48] which indicates the importance of the tRNA maturation in auxin signaling. Therefore, Elongator-mediated tRNA modification is involved in regulation of the PIN auxin transport protein levels which are required for proper auxin responses [48]. However, incomplete overlap of the Elongator and γ-toxin activities was observed as the *elp6 RP40p::gam* double mutant showed synergistic phenotype of delayed development in comparison to parental lines, suggesting for the Elongator complex and/or γ-toxin roles additional to the tRNA modification [48].

#### 4.2.4. Similar Role of Elongator and GRXS17

The activity of Elongator in the tRNA modification was further supported by the similarity between the *elo3-6* mutant and the *grxs17-1* line with defective expression of the *Arabidopsis* cytosolic monothiol glutaredoxin GRXS17, which associated in the tandem affinity purification assays with components of the cytosolic F-S assembly pathway and proteins implicated in the tRNA metabolism. GRXS17 might be required for transferring putative Fe-S clusters to these proteins, however, the exact role of this interaction is not investigated thoroughly yet [49]. GRXS17 binds cytosolic thiouridylase subunit1 and 2 (CTU1 and CTU2), which are both essential for the sulfur atom incorporation to the tRNA wobble uridine resulting in formation of mcm^5^s^2^U_34_ [49]. However, GRXS17 is not required for tRNA anticodon thiolation by the CTUs as evidenced by results of PAGE retardation assay visualizing thiolated tRNAs and showing no difference between the *grxs17-1* mutant and wild type Col-0. In line with the Elongator activity during the earlier stage of the same tRNA modification pathway, the *elo3-6* and *grxs17-1* mutants show very similar leaf and root phenotypes. Similarities were also noted at the molecular level as a large group of mostly induced genes involved in DNA-damage network overlapped between *grxs17-1* and *elo*/*elp* mutants [24,39,49]. Finally, at the physiological level, it was found that likewise the Elongator complex, GRXS17 takes part in the defense response against *B. cinereal* [44,49]. Interestingly, despite many similarities shared by the *grxs17-1* and *elo3-6* mutants, *grxs17* did not show a short hypocotyl phenotype of darkness-grown seedlings detected in *elo*/*elp* mutants representing the ELP1, ELP3, ELP4 subunits, and the DRL1 interactor of Elongator [43]. As shown in Figure 2, the hypocotyl lengths of the four-day-old darkness-grown seedlings of wild type Col-0 and the *grxs17-1* mutant in Col-0 ecotype are not significantly different in several independent experiments.

The short hypocotyl detected only in the *elo3-6* and not in *grxs17-1* mutants suggests that this phenotype is not a consequence of the compromised Elongator’s activity in tRNA modification and regulation of translation, but that it rather corresponds to another role of the complex, putatively played in the transcription regulation as indicated by transcriptome changes and reduced histone acetylation of selected genes in *elo3-6* [43].

#### 4.2.5. tRNA Wobble Uridine Modification Regulates Leaf Development

Recently, it was established that tRNAs with mcm^5^s^2^U_34_ modification are important for maintaining proper leaf morphogenesis [50]. Nakai and co-workers examined the Elongator-deficient plants and *urm11urm12* double mutants wherein the activity of URM1-like components of the wobble uridine sulfur modification pathway is compromised. Similarly to *elo*/*elp* mutants, *urm11urm12* completely lacked the sulfur modification shown in a gel retardation assays of sulfur modification of tRNAs. The phenotypes of *elo*/*elp* and *urm11urm12* were also comparable as these lines had expanded the leaf area, lower chlorophyll content, and under a stereomicroscope showed disordered mesophyll formation due to the increased intercellular spaces-with *elo*/*elp* phenotype being more severe. Furthermore, observation under scanning electron microscope indicated that the *elo3* and *urm11urm12* plants had increased the number of epidermal cells with parallel reduction of their size, which suggests endoreduplication cycling defects. Indeed, a delay in the second endoreduplication event was found in *urm11urm12*, whereas in the *elo3*, leaf cells exhibited delayed progression already at the first endoreduplication stage measured by flow cytometry analysis [50]. The *elo3* mutation enhances the leaf abaxial–adaxial polarity defect of the *as2* plants with mutation in the *AS2* (ASYMMETRIC LEAVES2) gene [94,95]. To explain whether the interaction between ELP3 and AS2 was caused by the Elongator function in the tRNA modification, leaf phenotypes of double *elo3as2* and triple *urm11urm12as2* mutants were compared. Only *elo3as2* exhibited the enhanced phenotype of trumpet/needle-like leaf shape and increased expression of genes involved in maintaining abaxial leaf polarity which was not found in *urm11urm12as2* [50]. These results suggest that the t-RNA modification does not affect the AS2 function, but rather another activity of Elongator—possibly the transcription-related activity—is linked to its function in abaxial–adaxial patterning of leaves.

The mcm modification mediated by Elongator is the prerequisite of the sulfur modification of wobble uridine [32,48]. Therefore, more severe phenotype of the Elongator-deficient plants than those lacking URM1-like proteins [50] might be a consequence of the absence of both the mcm and the sulfur modifications in *elo* mutants. The *urm11urm12* double mutant lacked only the sulfur modification [50]; and whether mutations in genes encoding components of the mcm^5^s^2^U_34_ modification pathway have influence on the level of mcm^5^U_34_ modification in plants is not defined yet. Moreover, corresponding abnormalities of the leaf development in *elo* and *urm11urm12* mutants indicate that the impairment of the tRNA wobble uridine modification leads to the delay of the endoreduplication cycling resulting in the unbalanced epidermal and mesophyll development [50]. Therefore, the Elongator-mediated tRNA modification together with tRNA thiolation are important for the fine-tuning the translation during leaf development.

In summary, the role of the Elongator complex in the tRNA wobble uridine modification in plants is confirmed, which allows to connect the Elongator activity with the regulation of translation. Thus far it was proven that Elongator-mediated tRNA modification translationally controls PIN proteins, and as a consequence, it regulates auxin responses which is crucial for maintaining proper plant development. Another factor which might be related to Elongator activity in tRNA wobble uridine modification is GRXS17 as its participation in the aforementioned process was proven; however, this participation is not investigated thoroughly yet. Finally, the most recently found processes wherein tRNA modification takes place are endoreduplication and leaf morphogenesis. Considering the above-given, it is interesting whether Elongator’s participation in the translation has any influence on plant processes which are not found yet. What is even more thought-provoking concerns the issue as to how translation activity is synchronized with transcriptional regulation of gene expression in plants because though in yeast the Elongator’s only activity is the regulation of translation, there is a substantial amount of evidence that supports a role for Elongator in the regulation of transcription in plants.

### 4.3. Elongator Acetylates alpha-Tubulin

Alpha and beta tubulins polymerase as heterodimers to form microtubules, of which functionality and dynamics are regulated by post-translational modifications including acetylation [96]. First experimental evidence showed that in mice, ELP3-as a component of the Elongator complex acetylates alpha tubulin [9]. In plants, ELP3 was purified together with PP2A phosphatase and histone deacetylase HDA14 upon microcystin-affinity chromatography, and all proteins were highly enriched in the microtubule fraction [10]. HDA14 interacts with the PP2A-A subunit and functions as a tubulin deacetylase, whereas ELP3 potentially plays an opposite role, namely it acts as an alpha tubulin acetylating enzyme [10] (Figure 1, Inset 4). Therefore, these results proved that ELP3 is able to acetylate non-histone proteins. However, the remaining subunits of the Elongator complex were not identified in this study, and therefore, it is not known whether ELP3 independently acetylates alpha tubulin or the fully assembled Elongator complex is required for maintaining this biochemical activity.

## 5. Do Subunits of the Plant Elongator Complex Always Act Together?

It has been well evidenced that all six ELP subunits are indispensable and have to cooperate as the integral complex to allow proper functioning of the plant Elongator. This conclusion has been repeatedly reached based on assays comparing morphological, physiological, and molecular features of plants harboring deleterious mutations in the various *ELP* genes and presenting very similar phenotypes. Additionally, the DRL1 interactor of Elongator has also been implicated in these studies, and in most cases, proved as necessary to assist the complex. The analyzed *elo*/*elp* mutants displayed highly related leaf phenotypes as well as gene expression profiles [24]; similar reductions in root growth [25,48,62]; common inflorescence, venation and root defects related to auxin biology [25], and comparably aberrant hypocotyl elongation in darkness or light [43]. The role of Elongator acting as the whole complex in enhancement of the leaf polarity defects associated with the mutation in *AS2* (*ASYMMETRIC LEAVES2*) was also suggested by a similar though not equally severe contribution of different *elo*/*elp* mutants to defective adaxial–abaxial polarity observed in the double *as2elo* mutants [95]. Mutations in each of *ELP1*-*ELP5* Elongator genes inhibit defense responses to bacterial pathogens [67], while all *elo*/*elp* mutants have reduced miRNAs accumulation, indicating that all Elongator subunits are required for miRNA biogenesis [45]. Finally, the wobble sulfur modification of tRNAs was equally affected in all analyzed *elo*/*elp* mutants, showing that the whole Elongator complex is involved in mcm modification, which is required for the subsequent sulfur modification [50]. Taken together, a number of strong lines of evidence linked both nuclear and cytoplasmic roles of Elongator to the complex configuration of its subunits. However, already in 2009, Zhou and co-authors discovered a substantial difference between the mutants representing various subunits of the Elongator complex [42]. Although all analyzed subunits were found similarly involved in the majority of the ABA response aspects, the stomatal closure was supersensitive to ABA only in *elp1* and *elp2* mutants, but not in *elp4* and *elp6*. Considering that ELP1 and ELP2 belong to the core subcomplex, while ELP4 and ELP6 are members of the accessory subcomplex, the differences in the stomatal movement observed between the mutants may result from different functions of the subcomplexes in the epidermal guard cells. However, the most spectacular results, challenging the concept of complex composition as the ultimate condition for all Elongator activities, were provided by heterologous overexpression of individual *Arabidopsis thaliana* subunits [69,70]. The *AtELP3* and *AtELP4* genes were introduced via Agrobacterium-mediated transformation into woodland strawberry *Fragaria vesca* [69]. Expression of *FvPR1* and *FvPR5*-orthologs of the *A. thaliana* defense genes regulated by Elongator was analyzed in lines overexpressing the AtELP3 or AtELP4 subunits. The transcript levels of both genes were significantly and markedly higher in the transgenic lines as compared to the control plants, indicating that the overexpression of subunits of the *Arabidopsis* Elongator leads to constitutive expression of the *FvPR1* and *FvPR5* genes. In line with high expression of the defense genes, the strawberry lines overexpressing AtELP3 or AtELP4 displayed enhanced resistance to fungal and bacterial pathogens. The same Elongator subunits were introduced into tomato *Solanum lycopersicum*, resulting in the increased—after being infected with *Pseudomonas syringae*—induction of the defense genes *PR1b1*, *PR-5x*, *DES* (*DIVINYL ETHER SYNTHASE*), and *ER1* (*ETHYLENE-RESPONSIVE PROTEASE INHIBITOR1*) as detected by RNASeq and confirmed via the qPCR assay [70]. In contrast to the strawberry transgenic lines, the overexpression of the AtELP3 or AtELP4 subunits in tomato plants did not lead to constitutive expression of the defense genes but only to the increased induction of these genes following the pathogen infection. However, similarly to strawberry transgenic lines, also in tomato overexpression of AtELP3 and, particularly, AtELP4 resulted in significantly enhanced resistance to tomato bacterial speck caused by *Pseudomonas syringae.* Similarly stunning consequences of the overexpression of individual Elongator subunits were earlier observed in human cells, wherein the ELP3 suppressed growth of 293T embryonic kidney cells and enhanced transcription, while ELP3 and ELP4 synergistically activated transcription [97]. In addition, in yeast, ELP3 suppressed the defects in the *anaphase-promoting complex 5* mutant [98]. As discussed in [69,70], effects of the individual subunits overexpression can be caused by activities of ELP3 and ELP4 independent from the Elongator complex or the *A. thaliana* ELP3/ELP4 may increase the activity of the Elongator complex in strawberry and tomato by an unknown mechanism. The second possibility is supported by the results of the complementation assays showing that the strawberry FvELP4-1 and the tomato SlELP3 and SlELP4 subunits are able to complement the respective *Arabidopsis thaliana elp3* and *elp4* mutants, indicating that the subunits of *A. thaliana*/strawberry or *A. thaliana*/tomato can interact and form functional complexes.

## 6. Conclusions

Studies of plant Elongator provide a large amount of data proving that, unlike in yeast, Elongator in plants combines several roles. The multifunctional role of the Elongator complex in plant cells is strongly evidenced by the nuclear and cytoplasmic localization of Elongator, which is in line with its biochemical activities specific for both cellular compartments. Therefore, in the nucleus, Elongator is involved in transcriptional regulation and DNA replication, and in the cytoplasm, the complex regulates translation and acetylates alpha tubulin (Figure 1 and Table 2). All activities of the plant Elongator are proved by convincing and high-quality assays including highly reliable mutant analyses which support both the translation- and transcription-related functions of the Elongator complex. The phenotypes of *elo/elp* mutants and *grxs17* or *urm11urm12* lines with defective expression of proteins involved in tRNA wobble U_34_ modification share many morphological, physiological, and molecular features, although show also some differences, indicating that Elongator plays an important role in tRNA modification, however, it has also other activities. On the other hand, strong resemblance between the *elo/elp* and SPT4-RNAi plants with affected expression of the transcription elongation factor SPT4 implicates that Elongator regulates transcription during the elongation phase. Therefore, in plants, the collected data strongly indicates the coexistence of transcriptional and translational activities of the Elongator complex and suggests their potential synchronization; however, the physiological role and mechanism of this interplay remain elusive. Contrary to the long-lasting and well-documented discussion about the dominant function of Elongator, data suggesting that individual subunits or sub-complexes may have activities independent of the entire Elongator complex are still sparse and mostly recent. Revealing the mechanism by which the heterologously overexpressed individual subunits of Elongator may effectively influence gene expression and pathogen resistance will be important to understand the exact mode of action of Elongator.

## Figures and Tables

**Figure 1 ijms-21-06912-f001:**
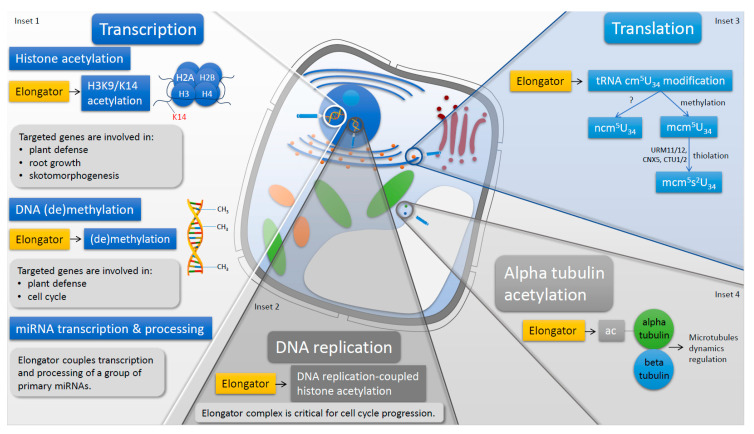
Schematic representation of the Elongator complex biochemical activities in a plant cell. In the nucleus, Elongator regulates the transcription through histone acetylation, DNA (de)methylation, and miRNA transcription and processing. The Elongator complex is involved in DNA replication which also occurs in the nucleus. As for cytoplasm, Elongator fine-tunes the translation by mediating tRNA wobble uridine modification and acetylates alpha tubulin.

**Figure 2 ijms-21-06912-f002:**
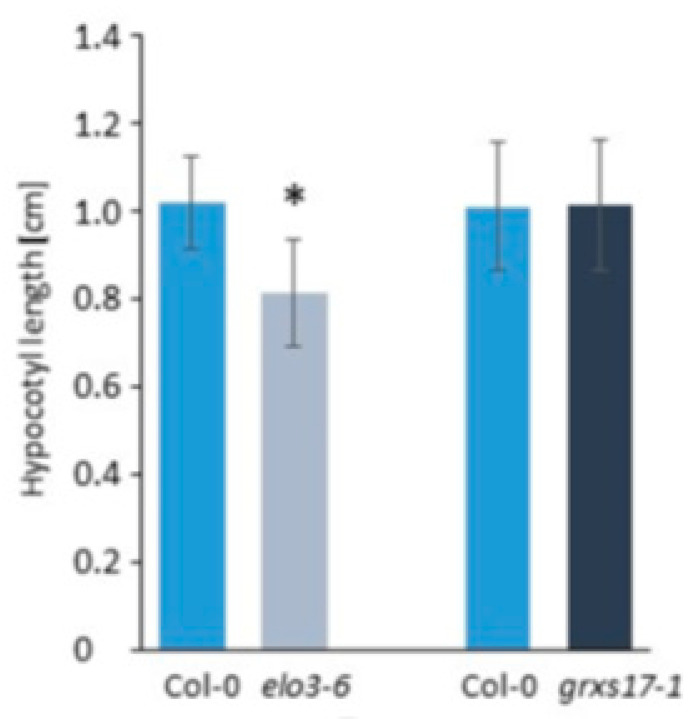
Hypocotyl lengths of 4-day-old *elo3-6*, *grxs17-1*, and their respective Col-0 wild-type control seedlings grown in darkness on half-strength MS medium using Image J analysis. Bars represent mean hypocotyl length of 30 seedlings (mean ± standard deviation). Differences between the mutant and wild type were statistically analyzed with an unpaired two-tailed Student’s t-test, * *p* < 0.01.

**Table 1 ijms-21-06912-t001:** Genes controlled by Elongator-mediated epigenetic regulation.

Epigenetic Regulation	Genes	Biological Process	Reference
Histone acetylation	*IAA3* and *LAX2*	Auxin signaling pathway	[25]
	*NPR1*, *PR2*, *PR5 EDS1*, *PAD4*	Plant defense	[40]
	*WRKY33*, *ORA59*, *PDF1.2*	Plant defense and ethylene signaling	[44]
	*PLT1*, *PLT2*, *SHR*, *SCR*, *PIN1*	Root development	[38,62]
	*RBOHD* and *ICS1*	Plant defense	[67]
	*LHY*, *HYH*, *HFR1*	Circadian rhythms and phytochrome signaling pathway	[43]
DNA (de)methylation	*NPR1* and *PAD4*	Plant defense	[40]
	*CYCB1*	Cell cycle	[38]

**Table 2 ijms-21-06912-t002:** Elongator regulates molecular mechanisms in the nucleus and the cytoplasm through different biochemical activities in *Arabidopsis thaliana*.

Localization	Molecular Process	Biochemical Activity	The Role of Elongator	Reference
Nucleus	Transcription elongation	Histone acetylation	Elongator regulates auxin signaling.	[25]
			Elongator regulates mitotic cell cycle and leaf patterning.	[39]
			Elongator is involved in plant defense.	[40]
			The role of Elongator in B. cinerea infection.	[44]
			Elongator regulates root development.	[38,62]
			Elongator is involved in plant defense.	[67]
			Elongator regulates skoto- and photomorphogenesis.	[43]
		-	IYO interacts with RNAPII and Elongator to promote elongation and initiate cell differentiation.	[71]
			Elongator and TEFs associate with elongating RNAPII.	[61]
			Elongator is involved in root stem cell maintenance through unknown mechanism.	[62]
	Transcription	DNA (de)methylation	Elongator is involved in plant defense.	[40]
			Elongator regulates root development.	[38]
		miRNA biogenesis	Elongator regulates miRNA transcription and processing	[45]
				[46]
		Unknown	Elongator regulates *MYBL2* expression which is involved in anthocyanin biosynthesis.	[42]
			Elongator mediates the establishment of leaf polarity through unknown gene expression system.	[95]
			Elongator acts as meristem cell cycle activator.	[75]
			Elongator controls expression of germination-related genes.	[63]
Cytoplasm	Translation	tRNA wobble uridine modification	Elongator-mediated tRNA modification is conserved in plants. Elongator is involved in mcm^5^s^2^U_34_ and ncm^5^U_34_ modification.	[32]
			Elongator and DRL1 are involved in ncm^5^U_34_ modification.	[47]
			Elongator regulates auxin responses by controlling PIN protein level.	[48]
			Strong resemblance between *elo3-6* and *grxs17*. GRXS17 is involved in tRNA modification.	[49]
			Elongator-mediated tRNA modification regulates leaf morphogenesis.	[50]
	Non-histone protein acetylation	Alpha tubulin acetylation	Elongator is involved in microtubules dynamics regulation.	[10]

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
