# Peer review of "Plant Elongator—Protein Complex of Diverse Activities Regulates Growth, Development, and Immune Responses"

_ijms, 2020, doi:10.3390/ijms21186912_

Round 1

Reviewer 1 Report

In this excellent review Jarosz and colleagues meticulously describe the molecular biology and the developmental function of elongator, a key protein acting in the elongation phase of transcription.

The authors resume the diverse molecular activities in both the nucleus and the cytoplasm of elongator in plants and describe in detail the different phenotypes of both loss and gain of function mutants.

I really enjoyed reading this review and I have only few comments.

L108 Elp6 mutant should be elp6

L154 Add the sigle BiFC for bimolecular fluorescence complementation

L156 Add the extended version of the names of HYL1, SE and DCL1

L156 As HYL1, SE and DCL1 are involved in miRNA biogenesis I would move this part in the 3.1.4 section and discuss the results in the light of miRNA biogenesis

L206 ABA should be reported in the extended version. Also, a sentence reporting that ABA is a plant hormone and that is involved in several activities in plants would help to clarify the reported results.

L569 Mutants combinations with AS2 are reported in this line, nonetheless AS2 is introduced only in line 621. Please correct and explain the rationale of the interaction of ELP with AS2.

Author Response

To Reviewer 1,

We are re-submitting our revised manuscript ID: ijms-931689 entitled: “Plant Elongator – protein complex of diverse activities regulates growth, development and immune responses” by Magdalena Jarosz, Mieke Van Lijsebettens and Magdalena Woloszynska. We highly appreciate the critical comments of the reviewers that were helpful to improve the quality of our article. All comments have been carefully addressed and the changes made in the manuscript are described in detail below and visible in the text in “Track Changes”.

Reply to Comments of Reviewer 1

Reviewer Comment (RC) 1: L 108Elp6 mutant should be elp6

Reply: L 113 in the revised manuscript – Elp6 was changed to elp6

RC 2: L154 Add the single BiFC for bimolecular fluorescence complementation

Reply: L 162 in the revised manuscript – BiFC was added

RC3: L156 Add the extended version of the names of HYL1, SE and DCL1

Reply: L 386 The extended versions of HYL1, SE and DCL1 were added and are now in the section 3.1.4 (following the next comment)

RC4: L156 As HYL1, SE and DCL1 are involved in miRNA biogenesis I would move this part in the 3.1.4 section and discuss the results in the light of miRNA biogenesis

Reply: L 385 – 387 The sentence concerning the interaction between Elongator and the components of the Dicing complex was moved to the 3.1.4 section.

RC5: L206 ABA should be reported in the extended version. Also, a sentence reporting that ABA is a plant hormone and that is involved in several activities in plants would help to clarify the reported results.

Reply 1: L 87 and 216 The abscisic acid name was added.

Reply 2: L 216 – 218 The short description of the ABA role in plants was added.

RC6: L569 Mutants combinations with AS2 are reported in this line, nonetheless AS2 is introduced only in line 621. Please correct and explain the rationale of the interaction of ELP with AS2.

Reply 1: L 601 in the revised manuscript – the extended version of the AS2 name was added.

Reply 2: L 601 – 604 – the rationale of the experiment and of the interaction of ELP3 and AS2 was explained and the following sentences (L 605 – 610) were changed accordingly.

Reviewer 2 Report

I enjoyed reading this informative review of Jarosz et al. about the interesting multifunctional Elongator protein complex in plants. In particular, I appreciate Figure 1, providing an overview over the manifold functions of this complex.

There is one point, however, that should be carved out more clearly. Elp3 is the catalytic subunit of the Elongator complex. Its C-terminal domain is a lysine acetyl transferase and its N-terminal domain is a radical SAM enzyme involved in the modification of uridine at the wobble position of tRNAs. In addition, Elp3 is supposed to possesses DNA methylation and/or demethylation activity. Is that really a proven fact or might the observed Elongator-dependent changes in DNA-methylation be of indirect nature as well? SAM-dependent DNA methyl transferases are normally characterised by a Rossman fold, which is utterly different from the fold of radical SAM enzymes. It seems very unlikely to me that the N-terminal domain of Elp3 is a radical SAM enzyme which, in addition, is able to catalyse both the methylation and demethylation of DNA.

Further points:

Line 63: Change “M.” to “Mus”, “C.” to “Caenorhabditis”, and “A.” to “Arabidopsis”.

Line 76: Change “D.” to “Drosophila”.

Line 80 to 82: The sentence “The elongata (elo) mutants in Arabidopsis thaliana (here “Arabidopsis” may be changed to “A.”), named for their elongated leaves, identified the Elongator subunits and therefore the Elongator mutants in plants are designated elo/elp” is hardly understandable. Do the authors mean something like that: “Elongata (elo) mutants of A. thaliana, named for their elongated leaves, led to the identification of the Elongator subunits and, therefore, Elongator mutants in plants are designated elo/elp”? The authors should change this sentence accordingly.

Line 155: Change “was showed” to “was shown”.

Lien 205: Change “… to regulate at the transcriptional level several response processes.” to “… to regulate several response processes at the transcriptional level.”

Line 313: “strengthens” should be changed to “corroborates” or “confirms”.

Line 326 to 328: “Histone acetylation decreased only in the coding region of the gene in the elo/elp mutant and not affected in the promoter implies that the coding sequence is targeted by Elongator for histone acetylation facilitating elongation of transcription, while …” does not make sense. It should be changed accordingly. In general, sentences which are that long should probably be avoided.

Line 477: Change “The lack of functional exchange was later …” to “Later, the lack of functional exchange was …”. Furthermore, “… was additionally confirmed indicating specific for species barriers or evolutionary diversification of Elongator and its regulatory proteins” does not make sense and should be corrected.

Ultimately, the authors obviously have problems how to use “the” or not correctly as well as in correct comma placement. The review should be revised in this regard.

Author Response

To Reviewer 2,

We are re-submitting our revised manuscript ID: ijms-931689 entitled: “Plant Elongator – protein complex of diverse activities regulates growth, development and immune responses” by Magdalena Jarosz, Mieke Van Lijsebettens and Magdalena Woloszynska. We highly appreciate the critical comments of the reviewers which were helpful to improve the quality of our article. All comments have been carefully addressed and the changes made in the manuscript are described below and visible in the text in “Track Changes”.

Reply to Comments of Reviewer 2

Reviewer Comment (RC)1: There is one point, however, that should be carved out more clearly. Elp3 is the catalytic subunit of the Elongator complex. Its C-terminal domain is a lysine acetyl transferase and its N-terminal domain is a radical SAM enzyme involved in the modification of uridine at the wobble position of tRNAs. In addition, Elp3 is supposed to possesses DNA methylation and/or demethylation activity. Is that really a proven fact or might the observed Elongator-dependent changes in DNA-methylation be of indirect nature as well? SAM-dependent DNA methyl transferases are normally characterised by a Rossman fold, which is utterly different from the fold of radical SAM enzymes. It seems very unlikely to me that the N-terminal domain of Elp3 is a radical SAM enzyme which, in addition, is able to catalyse both the methylation and demethylation of DNA.

Reply: The section 3.1.5 “Elongator modifies DNA methylation” has been rephrased and changed to address the comments of the reviewer:

  • L 396(1) The word “established”, related to the DNA demethylation activity of Elongator, has been changed to “described”.
  • L 396(2) The words “Elongator dependent”, referring to changes in DNA methylation, were removed.
  • L 397 The phrase “elo/elp mutants of thalianawas added for clarity.
  • L 400 – 402 and 404 – 407 were removed and replaced by L 402 – 404 to simplify the message and to emphasize difficulties in the interpretation of data concerning DNA methylation changes in the elo/elp mutants.
  • L 408 – 415 Single words or short phrases were removed for simplicity.
  • L 415 The phrase :Elongator is essential for” was changed to “Elongator is somehow involved in” to emphasize that the role played by Elongator in modification of DNA methylation is unclear.
  • L 417 The words: “demethylation, methylation or both” were removed for the same reason.
  • L 417 – 422 The fragment was added to explain that Elongator may indirectly modify DNA methylation and that the structure of the ELP3 SAM domain suggests that the Elongator complex may not be able to catalyze DNA methylation and demethylation.

Further points:

RC2: Line 63: Change “M.” to “Mus”, “C.” to “Caenorhabditis”, and “A.” to “Arabidopsis”.

Reply: L 66 and 67 in the corrected manuscript – the changes have been made.

RC3: Line 76: Change “D.” to “Drosophila”.

Reply: L 79 in the corrected manuscript – the change has been made.

RC4: Line 80 to 82: The sentence “The elongata (elo) mutants in Arabidopsis thaliana (here “Arabidopsis” may be changed to “A.”), named for their elongated leaves, identified the Elongator subunits and therefore the Elongator mutants in plants are designated elo/elp” is hardly understandable. Do the authors mean something like that: “Elongata (elo) mutants of A. thaliana, named for their elongated leaves, led to the identification of the Elongator subunits and, therefore, Elongator mutants in plants are designated elo/elp”? The authors should change this sentence accordingly.

Reply: L 83 - 85 in the corrected manuscript – the sentence has been changed accordingly.

RC5: Line 155: Change “was showed” to “was shown”.

Reply: L 164 in the corrected manuscript – “was showed” was removed (the whole sentence was rephrased and transferred to the section 3.1.4).

RC6: Line 205: Change “… to regulate at the transcriptional level several response processes.” to “… to regulate several response processes at the transcriptional level.”

Reply: L 215 in the corrected manuscript – the change has been made.

RC7: Line 313: “strengthens” should be changed to “corroborates” or “confirms”.

Reply: L 328 in the corrected manuscript – “strengthens” was changed to “corroborates”.

RC8: Line 326 to 328: “Histone acetylation decreased only in the coding region of the gene in the elo/elp mutant and not affected in the promoter implies that the coding sequence is targeted by Elongator for histone acetylation facilitating elongation of transcription, while …” does not make sense. It should be changed accordingly. In general, sentences which are that long should probably be avoided.

Reply: L 342 – 346 in the corrected manuscript – the sentence was removed.

RC8: Line 477: Change “The lack of functional exchange was later …” to “Later, the lack of functional exchange was …”. Furthermore, “… was additionally confirmed indicating specific for species barriers or evolutionary diversification of Elongator and its regulatory proteins” does not make sense and should be corrected.

Reply: L 506 – 508 in the corrected manuscript – the sentence was changed as suggested, additionally the phrase “indicating specific for species barriers or” was removed for clarity.

RC9: Ultimately, the authors obviously have problems how to use “the” or not correctly as well as in correct comma placement. The review should be revised in this regard.

Reply: The review has been revised by professional translator. All changes in the manuscript, which has not been mentioned and described above, were made by translator.